# Development and validity evidence for the intraprofessional conflict exercise: An assessment tool to support collaboration

Nadia M. Bajwa[1,2]*, Julia Sader[2], Sara Kim[3], Yoon Soo Park[4], Mathieu R. Nendaz[2], Naïke Bochatay[5]

1 Department of General Pediatrics at the Children's Hospital, Geneva University Hospitals in Geneva, Geneva, Switzerland, 2 Unit of Development and Research in Medical Education (UDREM), Faculty of Medicine, University of Geneva, Geneva, Switzerland, 3 Department of Surgery, School of Medicine, University of Washington, Seattle, Washington, United States of America, 4 MGH Institute of Health Professions at Massachusetts General Hospital, Boston, Massachusetts, United States of America, 5 Department of Pediatrics at the University of California, San Francisco, San Francisco, California, United States of America

* Nadia.Bajwa@hcuge.ch

**Data Availability Statement:** "All data files are available from the Yareta database (https://doi.org/10.26037/yareta:ar2yfvbqfrhn7pvzlilno4uvym)."

## Abstract

### Background

Effective collaboration is the foundation for delivering safe, high quality patient care. Health sciences curricula often include interprofessional collaboration training but may neglect conflicts that occur within a profession (intraprofessional). We describe the development of and validity evidence for an assessment of intraprofessional conflict management.

### Methods and findings

We designed a 22-item assessment, the Intraprofessional Conflict Exercise, to evaluate skills in managing intraprofessional conflicts based on a literature review of conflict management. Using Messick's validity framework, we collected evidence for content, response process, and internal structure during a simulated intraprofessional conflict from 2018 to 2019. We performed descriptive statistics, inter-rater reliability, Cronbach's alpha, generalizability theory, and factor analysis to gather validity evidence. Two trained faculty examiners rated 82 trainees resulting in 164 observations. Inter-rater reliability was fair, weighted kappa of 0.33 (SE = 0.03). Cronbach's alpha was 0.87. The generalizability study showed differentiation among trainees (19.7% person variance) and was highly reliable, G-coefficient 0.88, Phi-coefficient 0.88. The decision study predicted that using one rater would have high reliability, G-coefficient 0.80. Exploratory factor analysis demonstrated three factors: communication skills, recognition of limits, and demonstration of respect for others. Based on qualitative observations, we found all items to be applicable, highly relevant, and helpful in identifying how trainees managed intraprofessional conflict.

**Funding:** The author(s) received no specific funding for this work.

**Competing interests:** The authors have declared that no competing interests exist.

## Conclusions

The Intraprofessional Conflict Exercise provides a useful and reliable way to evaluate intraprofessional conflict management skills. It provides meaningful and actionable feedback to trainees and may help health educators in preparing trainees to manage intraprofessional conflict.

## Introduction

To become physicians, medical trainees must demonstrate that they are competent in both collaboration and communication [1–3]. Competency in collaboration requires medical trainees to develop skills in negotiation and conflict management [4]. To date, research and educational initiatives have mostly addressed these skills in the context of interprofessional education, where trainees from different health professions have an opportunity to "learn about, from and with each other" [5]. These skills need to be taught progressively, in a developmentally appropriate manner [6]. This is coupled with the professional identity formation of medical trainees over time and transitions into new roles [7–9]. While senior trainees who have a stronger professional identity may benefit from educational opportunities focusing on interprofessional negotiation and conflict management, junior trainees may find it challenging [10]. Addressing these skills *intraprofessionally* may better support development of collaborative competencies and, ultimately, prepare trainees to work more effectively as part of a healthcare team [11, 12].

By first learning communication skills to deal with conflicts intraprofessionally, trainees have the opportunity to lay down the foundation for conflict management skills before adding the additional complexity that may arise in interacting with other professions. Ignoring the importance of collaboration can lead to medical errors that negatively impact patient care. The Joint Commission demonstrated that human factors, communication, and leadership difficulties were often times the primary cause of sentinel events [13]. Conflict management curricula must prepare trainees to deal with the inevitable conflict that may arise between colleagues in the clinical setting where channels of constant communication and negotiation are ubiquitous and in an environment where there may be poorly understood roles amongst professionals, power differentials, and a lack of systems and training [14, 15].

Intraprofessional conflicts differ from interprofessional conflicts on multiple fronts. Foremost, they make up a larger proportion of conflicts that occur in the workplace compared to interprofessional conflicts [16, 17]. They are most often linked to poor relationships and are dealt with by using avoiding and forcing strategies that may not generate a constructive solution to the conflict [18]. Factors contributing to poor interpersonal relationships are a lack of time, inappropriate communication, and resentment stemming from past conflicts [19]. Intraprofessional conflicts tend to spur professionals to consider leaving their professional position and may impact staff turnover [18]. They are more likely to impact individuals at a personal level by leading to emotional distress and burnout [18]. Because intraprofessional conflicts typically occur between close co-workers as opposed to more distant members of the healthcare team, intraprofessional conflicts are more likely to be dealt with using the avoiding or forcing/dominating response instead of more effective responses such as the integrating or compromising response [18]. The occurrence of intraprofessional conflict may be explained by social identity theory where a group of physicians may identify them as the "in-group" and consider those physicians that do not share their same characteristics as the "out-group" leading to a negative view of the other group [20, 21]. For example, physicians who identify

themselves as belonging to one subspecialty may look down upon physicians who work in primary care. Preparing trainees to manage intraprofessional conflicts could lessen the impact of these conflicts by rendering trainees more resilient and skilled in addressing intraprofessional conflict constructively.

Effective intraprofessional collaboration is seldom role-modeled or discussed explicitly in clinical settings, making it challenging for trainees to identify constructive conflict management strategies when disagreements occur between colleagues. Rendering intraprofessional collaboration as an explicit competency has the potential to support trainees' development of broader collaboration and communication skills. The literature now supports the need to create models for intraprofessional collaboration based on team-based science that are considered as important as the models that exist for interprofessional collaboration [21–23]. However, existing literature currently lacks descriptions of effective intraprofessional curricula that could support educators. There is a need to inform the development of such curricula in order to facilitate the transfer of knowledge and competence about intraprofessional collaboration to medical trainees.

Assessing collaboration and communication is complex as it needs to incorporate individual and team behaviors in different contexts [24] and can create tensions between trainees and educators [25]. This study describes the development of and validity evidence for an assessment of trainees' competence in intraprofessional conflict management skills. Basing the instrument on behaviors that have previously been shown to be effective in managing intraprofessional conflict may allow for this assessment tool to be integrated into conflict management training.

## Methods

### Instruments and variables

The purpose of the Intraprofessional Conflict Exercise (ICE) is to observe and measure the behaviors that trainees demonstrate during the management of an intraprofessional conflict. The 22 item ICE is based on our review of the literature on conflict management in health care and beyond, including the concepts of principled negotiation, the DESC method (describe, express, specify, and consequences), the principles of non-violent communication, the Can-MEDS collaborator role, and the Professionalism Mini-Evaluation exercise [1, 26–28]. We included a global evaluation at the end of the scale. The scale for each item ranges from zero to four: (0) not applicable, (1) unacceptable, (2) below expectations, (3) meets expectations, and (4) exceeds expectations. For each item, the participant's overall performance is rated. Frequency counts of each behavior are not meant to be collected. Table 1 lists the items and S1 Table includes the full instrument. Two physician educators drafted the instrument items and then a sociologist and an educational specialist revised it to improve interpretation of the items. We piloted the final version of the instrument during the training sessions with the standardized colleagues and did not deem that further changes were necessary. A schematic representation of the ICE development process is described in S1 Fig.

### Data collection

We collected data over a two-year period from 2018 to 2019, as part of a simulation assessment in our pediatric residency admissions process. We recruited a convenience sample of participants from the applicants to the Geneva University Hospitals Pediatric Residency Program. Based on their curriculum vitae, personal statement, exam scores, and medical school attended (preference for Swiss medical school graduates), applicants are invited to an interview day. During this day they undergo one structured behavioral interview with two members of the

**Table 1. Qualitative observations of the intraprofessional conflict exercise items.**

| Item | Observations of applicants' use of each item in the context of a simulated intraprofessional conflict | Illustrative quote from observation notes |
|---|---|---|
| 1. Listened actively to colleague | Used by all applicants, typical behaviors include nodding and looking at the SP while they speak. This item was particularly relevant to this scenario as the SP was not replying to emails and was purposely ignoring the resident, therefore this strategy was actively used to reinstate communication. | *The applicant enters the room, the SP says: "Great to finally see you." The applicant looks at the SP and apologizes: "Yeah, I'm sorry, I've been having a hard time responding to emails." (A17)* |
| 2. Showed interest in the colleague as a person | Used by most applicants. This item encompasses different ways by which the applicant expresses their interest to help dissolve the tension. | *"I'm happy to work on the slides! I can imagine you will be tired at the end of your week of night shifts." (A6)* |
| 3. Identifies nonverbal cues from colleagues | Used by some applicants, this item correlates strongly with item 1 (listened actively to colleague). | *The applicant says: "We have one week, so that's a lot of time." Looks at the SP and asks notices that she seems unhappy: "Do you find that stressful? Is this difficult for you?" (A15)* |
| 4. Demonstrates empathy for their colleague | Used by most applicants. Some applicants demonstrated empathy and apologized or showed understanding, others were more defensive. | *The applicant asks the SP how many nights she will be working and shows empathy: "Yeah, that's really not a good week for you to have to work on the journal club presentation. . ." (A14)* |
| 5. Uses "I" statements to effectively communicate their emotions and point of view | Used by all applicants. | *"Usually, if we both worked on a project, I like to have both people present, just to show that we worked together." (A3)* |
| 6. Identifies the needs or interests of the colleague | Used by some applicants to create a bond or try to mend and repair the relationship. | *"Are you more interested in one of these articles? Have you seen anything like that on your rotations, do you think you would have a scenario for either of these?" (A6)* |
| 7. Describes the situation in terms of a shared observation | Very rarely used by applicants; when used, shows the applicant's reflective behavior. | *"Oh, you're an intern as well? I thought you had more experience." (A14)* |
| 8. Admitted errors/omissions | Used by some applicants, usually in response to a comment by the SP. | *The applicant apologizes and says that she has not had time to look at what they have to do, she asks the SP if she knows how to proceed. (A2)* |
| 9. Maintained composure in a difficult situation | Used by most applicants, (e.g., A 20), although a few (e.g., A15) got angrier | *"I don't see how this can work out honestly, you have not done any of the work." (A15)* |
| 10. Addressed own gaps in knowledge and skills | Used by some applicants, usually as a strategy to address the conflict. | *"Okay, well, I was not sure either on the instructions so maybe if we can find out what we need to do, we can prepare a great presentation." (A20)* |
| 11. Negotiates a mutually agreeable solution to the conflict | Used by most applicants. Out of all of the items this item was highly present during the encounter and seemed to bring resolution to most of the conflicts or problems discussed during the simulation. | *"How about we talk about who presents what later? For now, I'll work on the slides, then we analyze the article together, and you could perhaps work on the clinical case?" (A9)* |
| 12. Verbalizes specific solutions to the conflict | Used by some applicants, typically towards the end of the simulation and observable through suggestions of key solutions to resolve the conflict. | *"Okay, so in summary, you will send the clinical case to everyone tonight, and I'll start working on the slides. I will email you this weekend to let you know what I have, and we go from there." (A14)* |
| 13. Uses nonverbal communication to convey openness and supportiveness | Used by most applicants, behaviors included open arms, nods with the head and tilting the head to show understanding. | *The applicant shows a lot of understanding, looks at the SP, immediately takes the blame when the SP complains about her lack of communication. (A31)* |
| 14. Demonstrated awareness of limitations | Used by some applicants, often to discuss external constraints rather than personal limitations. | *"I'm an intern so this is my first journal club, whereas you already have experience with that." (A1)* |
| 15. Solicited feedback | Used by some applicants, often in combination with item 7 (Identifies the needs or interests of the colleague). This item was used to ensure that the colleague agrees with next steps. | *"I could work on a draft for the slides and send them to you this weekend so you can take a look. Would that be okay?" (A15)* |
| 16. Accepted feedback | Rarely used by applicants; when used, typically through comments to show acknowledgement of the colleague's feedback to ensure effective resolution of the conflict. | *"I hear what you are saying, maybe next time I should call you instead of sending you emails" (A2)* |
| 17. Maintained appropriate boundaries | Rarely used by applicants. | *"Since you're so busy, I'm happy to do a little more, but what can you do in exchange?" (A17)* |
| 18. Maintained appropriate appearance | Used by most applicants. | *The applicant's facial expression shows that she disagrees with the SP's proposition as she respectfully says: "I honestly think that they expect us to work on this together." (A7)* |

*(Continued)*

**Table 1.** (Continued)

| Item | Observations of applicants' use of each item in the context of a simulated intraprofessional conflict | Illustrative quote from observation notes |
|---|---|---|
| 19. Demonstrated respect for colleagues | Used by all applicants | *"Thank you for meeting with me, I can understand where you are coming from, I just hope we can work together on this (…) thank you I appreciate it.* (A6) |
| 20. Avoided derogatory language | Used by most applicants. The tone stayed composed and contained. | *"I am just frustrated because I wanted to be done by now, and it does not leave us a lot of time"* (A16) |
| 21. Avoids verbal attacks on the colleague | No verbal attacks were observed. | |

admissions committee, provide two standardized letters of recommendation, and participate in three simulated cases observed by two examiners per case [29]. Most applicants completed medical school at the University of Geneva where communication skills training includes but is not limited to conducting an interview with a patient, breaking bad news, and conducting handoffs. Conflict management is not specifically addressed. The use of the ICE between undergraduate and graduate medical education was appropriate given applicants' likely previous exposure to training in communication. While the use of the ICE is limited in our study to a residency admissions process, the ICE is intended to be used in a variety of settings as part of conflict management training at any stage of the health professions education continuum.

One of the three simulated cases focuses on a conflict with a standardized colleague (SC), an actor/actress playing the role of a colleague. We chose to have the case take place in the post-graduate setting to simulate a workplace setting. We designed the case to cover all items of the ICE and to last 13 minutes. In this case, the applicant experiences a conflict with the SC while preparing for a journal club. The SC was instructed to be confrontational and to seek to convince the applicant to do all the work. A patient care situation was not included in the case as the two other cases already involved patient care situations and because intraprofessional conflicts can arise for various reasons beyond patient care disagreements [18]. Two professional trainers from the simulation center trained the SCs and the case was piloted with volunteer residents. Two trained pediatric faculty examiners observed and rated each applicant using the ICE. We trained examiners to use the ICE prior to the simulation by having them observe a video recording of the SC case and by practicing filling in the instrument. Examiners were specifically briefed on their understanding of the ICE items. A discussion with the examiners was used to develop consensus about what an appropriate response to the situation would be. On the day of the simulation, raters observed each participant through a two-way mirror.

On the day of the simulation, one of two qualitative researchers (a sociologist or a psychologist, depending on the day) observed the SC encounters. As the qualitative researchers were involved in the creation of the ICE, they did not undergo rater training. They used the ICE as a semi-structured observation guide and took notes on how each item manifested itself in the encounter. They also paid particular attention to interactions and processes at play during the simulated conflict. Before entering the simulation, participants were introduced to the scenario with a written introduction giving context to the case and specific tasks to be performed (review the instructions for the journal club with their colleague, choose an article to be presented together, and decide on the distribution of labor for the preparation of the presentation). Participants were not debriefed immediately after the simulated exercise and scores were not immediately shared with the participants. Participants were invited after the admissions decision to a voluntary debriefing session where a video of the performance was viewed, scores were shared, and constructive feedback was given to each participant.

The Institutional Review Board at the Geneva University Hospitals (September 5, 2018) and the University of Illinois at Chicago (October 25, 2018, Research Protocol # 2018–1167) granted an exemption for ethical approval for this study. We obtained written informed consent to analyze applicants' de-identified admission data from all participants.

## Data analysis

In this study we used a concurrent triangulation mixed methods design to gather validity evidence for the ICE [30]. Validity evidence for the ICE examined the following Messick's unified sources of validity evidence: content, response process, and internal structure as operationalized by Downing and in the *Standards for Educational and Psychological Testing* [31, 32].

1. Content- We collected content validity evidence for the ICE through the identification of items relevant for the management of intraprofessional conflict through an extensive literature search. Common themes in intraprofessional conflict were mapped to the items on the ICE and to the case to ensure that most items would be pertinent to the case. A panel of faculty members with expertise in conflict management evaluated the appropriateness of the items that are included in the scale. We calculated descriptive statistics for each of the items in the ICE and the average number of items rated to determine the representativeness of the items in the scale. Alpha level was set to .05.

2. Response Process- Response process was determined through a qualitative approach. During the piloting of the ICE, cognitive pretesting was conducted with potential examiners (interviewers were questioned on their interpretation and thought process related to each item) to better understand the cognitive validity of scale items [33]. Ambiguous items were modified to improve comprehension. During the simulation, the two qualitative researchers documented the appropriateness of the items on the scale, the presence of behaviors that are not accounted for in the scale and identified potential sources of construct irrelevant variance. Together with the first author, the qualitative researchers used a directed approach to content analysis to compare their observation notes and analyze them [34]. They used Microsoft Excel 16 for Macintosh for the analysis of qualitative data (Microsoft Corporation, Redmond, Washington).

3. Internal Structure- We examined four sources of internal structure validity evidence: (1) inter-rater reliability, (2) Cronbach's alpha, (3) factorial structure of the ICE, and (4) variance components analysis (Generalizability study). We used quadratically weighted kappa to evaluate inter-rater reliability. We conducted an exploratory factor analysis (EFA) using Maximum Likelihood with oblique Promax rotation to better understand the factor structure of the ICE. We used multivariate imputation for missing values. In addition, we performed generalizability and decision studies to estimate variance components and reliability indices of ICE scores [35]. Applicants ($P$) are the objects of measurement, with three facets (sources of error variance): raters ($R$), and items on the ICE instrument ($I$). We analyzed different sets of fully-crossed data using the $P$ X $R$ X $I$ design [35]. All facets were assumed to be random samples from the universe of potential raters and items; this assumption allows for more conservative baseline estimates of reliability for this novel SC-based ICE assessment. Estimation of variance components were conducted using urGenova (Brennan, 2001); urGenova employs estimation of variance components with missing data. We performed all quantitative analyses with StataSE 16 for Macintosh (Stata Corp, College Station, Texas).

## Results

We collected 164 observations (2 rating forms per applicant) for 82 applicants over the 2-year study period. 62 (76%) of the applicants were women and 16 (20%) of applicants completed their medical school studies outside of Switzerland.

### Content

The average item score (M, SD) was 3.18 (0.45). The average number of completed items (M, SD) was 19.81 (1.71). Items most likely to be noted not applicable were "admitted errors/omissions", "solicited feedback", and "accepted feedback". The following items were more likely to be noted "below expectations": "identifies nonverbal cues from colleague", "identifies the needs or interests of the colleague", "describes the situation in terms of a shared observation", and "negotiates a mutually agreeable solution to the conflict".

### Response process

Based on our qualitative observations of the simulated encounters, we noted that applicants used similar strategies to try and resolve the conflict and demonstrated most behaviors included in the ICE. Applicants first sought to understand the SC's perspective, which led them to listen actively (item 1), to show interest in the SC (item 2), to demonstrate empathy (item 5), and to be respectful of the SC (item 17). When the applicants realized that the SC was not cooperating, they tried to mitigate the conflicts by using "I" statements (item 6), trying to identify the needs of the SC (item 7), highlighting their common goals (item 9), and negotiating a solution (item 20). Some applicants placed themselves in a vulnerable position by discussing their own mistakes and limitations (items 8, 10, and 16), and by soliciting and accepting feedback from the SC (items 11 and 12). At the end of the encounter, some applicants concluded with a summary of the resolution (item 21). Throughout the encounter, most applicants demonstrated openness, active listening, and maintained appropriate appearance and language (items 3,4, 13, 14, 15, 18). Some applicants avoided the conflict and let the SC take the lead in the interaction as demonstrated in this quote:

> The applicant says: *"So in summary, you'll send an email to the attendings to share our papers for the journal club. I'll work on the power point, you give me feedback, and then I'll present?"* *The applicant says that he agrees with the SC that it is a good idea if the person who worked on the slides presents them. He shows empathy for the SC who had said that she will be on call overnight and will be tired afterwards and says that presenting will be good practice for him. He concludes*: *"That sounds great! Working with you is interesting, thank you." (A6)*

In contrast, other applicants tried to negotiate a solution that they believed would be fairer than what the SC proposed:

> *"I don't know how much time you spent looking for articles. . . It's not that I don't trust you, but I feel as though you're dumping all the work on me. I'm not really comfortable with this way of working together." (A15)*

We provide illustrative quotes and observations in Table 1.

### Internal structure

**Reliability.** Inter-rater reliability was fair with a weighted kappa of 0.33 (SE = 0.03). Cronbach's alpha for the scale was 0.87.

**Table 2. Exploratory factor analysis of the ICE: Factor loadings based on rotated factor matrix solution.**

| | Factor | | |
|---|---|---|---|
| Item | 1 Communication Skills | 2 Recognition of limits | 3 Demonstration of respect for others |
| Listened actively to colleague | **0.482** | -0.009 | 0.341 |
| Showed interest in the colleague as a person | **0.576** | -0.092 | 0.064 |
| Identifies nonverbal cues from colleagues | **0.460** | -0.149 | -0.039 |
| Demonstrates empathy for their colleague | **0.738** | -0.110 | 0.028 |
| Uses "I" statements to communicate effectively their emotions and point of view | **0.349** | 0.096 | 0.258 |
| Identifies the needs or interests of the colleague | **0.763** | 0.021 | -0.173 |
| Describes the situation in terms of a shared observation | **0.472** | 0.182 | -0.006 |
| Admitted errors/omissions | **0.404** | 0.335 | -0.039 |
| Maintained composure in a difficult situation | **0.362** | 0.191 | 0.107 |
| Addressed own gaps in knowledge and skills | **0.448** | -0.172 | 0.275 |
| Negotiates a mutually agreeable solution to the conflict | **0.743** | 0.115 | -0.120 |
| Verbalizes specific solutions to the conflict | **0.521** | 0.326 | -0.145 |
| Global evaluation | **0.819** | -0.035 | -0.065 |
| Uses nonverbal communication to convey openness and supportiveness | **0.369** | **0.369** | -0.003 |
| Demonstrated awareness of limitations | 0.238 | **0.283** | 0.124 |
| Solicited feedback | -0.047 | **0.817** | -0.035 |
| Accepted feedback | -0.006 | **0.955** | 0.089 |
| Maintained appropriate boundaries | -0.045 | 0.133 | **0.666** |
| Maintained appropriate appearance | -0.175 | 0.329 | **0.648** |
| Demonstrated respect for colleagues | **0.325** | 0.000 | **0.382** |
| Avoided derogatory language | -0.146 | -0.032 | **0.996** |
| Avoids verbal attacks on the colleague | 0.046 | -0.062 | **0.821** |

Extraction method: Maximum Likelihood. Rotation method: Promax with Kaiser normalization.

**Factorial structure.** The EFA revealed three factors based on the scree plot and Eigenvalue >1. The KMO was 0.84 and Bartlett's test of sphericity was significant indicating that the data was suitable for factor analysis. We interpreted the three factors as factor 1 communication skills, factor 2 recognition of limits, and factor 3 demonstration of respect for others. 53% of the total variance was attributed to communication skills, recognition of limits accounted for 26% of the variance, and demonstration of respect for others accounted for the remaining 21% of variance. Certain items loaded to two factors such as "uses nonverbal communication to convey openness and supportiveness" which loaded to both communication skills and recognition of limits. "Demonstrated respect for colleagues" also loaded to both communication skills and demonstration of respect for others. Factor loadings for each of the items are reported in Table 2.

**Variance components.** We analyzed the internal structure of the ICE by grouping the results from the two study years. The object of measurement (applicant) variance (19.7%) demonstrated the ability of the ICE to effectively differentiate among applicants. The highest proportion of variance came from the interaction of applicant with the item (11.0%); see Table 3. The ICE with only one case and two raters demonstrated high reliability with a G-coefficient of 0.88 and Phi-coefficient of 0.88. The decision study showed that it is possible to use only one rater and still have a high reliability, G-coefficient 0.80.

**Table 3. Generalizability study results from the 2018 and 2019 combined.**

| Effect | Variance Component | %Variance Component |
|---|---|---|
| Learner (P) | 0.037 | 19.7% |
| Rater (R) | 0.000 | 0.0% |
| Item (I) | 0.002 | 0.8% |
| Learner-Rater Interaction (P X R) | 0.002 | 1.3% |
| Learner-Item Interaction (P X I) | 0.020 | 11.0% |
| Rater-Item Interaction (R X I) | 0.006 | 3.5% |
| Residual Error | 0.118 | 63.7% |

ICE data (n = 82 applicants, 164 forms)

## Discussion

The Intraprofessional Conflict Exercise assesses three distinct domains related to conflict management: communication skills, recognition of limits, and demonstration of respect for others. The construct validity of the scale indicates that it can be used to provide meaningful and actionable feedback to applicants. Validity evidence gathered on the ICE based on the internal structure shows high reliability with the ability of the ICE to differentiate between applicants based on their performance on one case making the ICE suitable for high-stakes assessments as well as formative assessments. The generalizability study demonstrated that applicant performance varied significantly based on the item rated indicating that item difficulty varied. Response process evidence revealed that certain items related to active listening were related with the overall performance. Observations of the performance also showed which items were used more frequently and how applicants may have used different strategies to demonstrate their competency related to the item. Items that were most likely to be noted "below expectations" were related to identifying the perspective of the colleague and choosing a strategy to manage the conflict. Focusing feedback and further training specifically on these items may inform and complement conflict management training.

A strength of our study is the reliable performance of the ICE in a high-stakes assessment such as an admissions process. The established validity evidence lays the groundwork for and supports future use of the ICE as a formative assessment in a conflict management curriculum. Creating rigorous assessments related to intraprofessional collaboration is a first step in complementing the current conflict management curricula that exist in medical schools and postgraduate training programs. Conflict management curricula are often based on an ideal vision of collaboration in the workplace and may not pay sufficient attention to the complexities of resource management and role negotiation that are often times a source of conflict [4, 36]. Using simulation to practice these conflict management skills allows participants to actively practice their competencies and to receive pertinent and reliable feedback on their performance. The three domains of the ICE (communication skills, recognition of limits, and demonstration of respect for others) may already be taught in other communication or teamwork curricula; However, contextualizing the use of these skills in the setting of conflict allows for a synergistic application of these skills. The advantage of using the ICE in simulation is that it allows learners to step out from a theoretical silo where responses to situations are predictable and permits them to respond to complex situations in a safe environment. Using the ICE as a formative assessment, may allow for the measurement of progression of competence on entrustable professional activities related to collaboration or may provide a tool for remediation for trainees experiencing difficulty with collaboration [37]. Direct supervision using the ICE may allow for focused feedback on items that may be more challenging for learners.

Ultimately, interventions that aim to enhance collaboration have been shown to improve the efficiency of clinical processes and patient health outcomes [38].

This assessment was specifically formulated to address intraprofessional collaboration, a critical competency that may be taught and evaluated as a first step to developing collaborative competencies both intra- and interprofessionally [12]. Other health professions such as nursing share contextual and cultural similarities to physicians. We believe that the items in the ICE may be generalizable to other health professions and that future studies testing the ICE with other health professions may contribute to the literature on intraprofessional conflict. Future studies may also adapt the ICE to include items related to address differences in roles and hierarchy that may allow for the application of this assessment in an interprofessional context. While our study was also conducted in a simulated setting, which allows for consistent observations across applicants, the ICE may also be relevant to the clinical setting. Further research using the ICE as a workplace-based assessment may contribute to the validity evidence for the assessment.

The assessment of collaboration must be coupled with strategies that favor productive intraprofessional collaborative practices. For example, taking advantage of the learning that takes place during informal professional interactions is an opportunity to deliberately integrate reflection and to optimize learning [39]. Collaboration has been shown to be positively influenced by communication, respect, professionalism, the climate of collaboration, and quality of care [40]. Therefore, targeting conflict management training to focus on communication skills may have the power to shift responses to conflict and lead to resolution. Focusing intraprofessional education on dealing with mindset, professional identity formation, and power dynamics is an effective way of circumventing barriers to establish an environment that is favorable towards intraprofessional collaboration [41]. Finally, power dynamics can be influenced by limiting hierarchy, reducing inequity, and by supporting psychological safety so that physicians feel capable of speaking up when disagreements occur [42, 43].

Despite the rigorous methods used to gather validity on the ICE, our study has limitations. While the simulation was high fidelity, the complexity of both the verbal and non-verbal interactions between the applicant and the SC may not have been fully captured by the raters. Certain sentiments of applicants such as frustration or empathy may not have been explicit, and a debriefing of the case would have been helpful to better understand the applicant's understanding of the case and their reactions. The four-point scale that we chose to use in our study may have limited the interpretation of the generated scores as there was no predetermined performance standard for our case; testing the possibility of using a Likert frequency scale may give further insight into the competency of our participants. We used the ICE with one case scenario focused on a professional development task. The study of the use of the ICE with various intraprofessional collaboration challenges, specifically those concerning patient care, would add to the ICE's validity evidence. More specifically, we would like to see the application of the ICE in an undergraduate teaching setting on conflict management with the goal of preparing students for the workplace. Our study was also conducted in one specialty at one European institution. Interpretation of our findings therefore needs to take this context into account and future studies may focus on other specialties and cultural contexts. In addition, participants were not questioned on their previous communication skills or conflict management training, and this may have had an impact on their performance. Finally, while the portrayal of the case was standardized amongst the SCs there is the possibility that the different portrayals may have influenced the reactions of the applicants. While little variance was attributed to raters in the generalizability study, implicit biases may have also contributed to bias in the evaluations given by the raters. The development of future cases to use with the ICE will contribute to the generalizability of our results.

## Conclusions

Intraprofessional conflict is a regular occurrence in the clinical setting. Training medical students and residents to respond to these conflicts in a constructive manner contributes to the ability of the healthcare team to create an environment that is ideal for patient care. Rigorous assessments that provide meaningful and actionable feedback to trainees may supplement current conflict management training for medical trainees.

## Supporting information

**S1 Fig. The intraprofessional conflict exercise development process.**
(DOCX)

**S1 Table. Example of the intraprofessional conflict exercise.**
(DOCX)

## Acknowledgments

The authors wish to thank the applicants and faculty members that participated in this study for their time and collaboration.

**Previous presentations:** Poster presentation Association for Medical Education in Europe, 2020.

## Author Contributions

**Conceptualization:** Nadia M. Bajwa, Julia Sader, Sara Kim, Yoon Soo Park, Mathieu R. Nendaz, Naïke Bochatay.

**Data curation:** Nadia M. Bajwa, Julia Sader, Yoon Soo Park, Naïke Bochatay.

**Formal analysis:** Nadia M. Bajwa, Julia Sader, Yoon Soo Park, Naïke Bochatay.

**Investigation:** Nadia M. Bajwa, Julia Sader, Naïke Bochatay.

**Methodology:** Nadia M. Bajwa, Julia Sader, Sara Kim, Mathieu R. Nendaz, Naïke Bochatay.

**Project administration:** Nadia M. Bajwa.

**Resources:** Nadia M. Bajwa.

**Supervision:** Sara Kim, Yoon Soo Park, Mathieu R. Nendaz.

**Validation:** Nadia M. Bajwa.

**Writing – original draft:** Nadia M. Bajwa, Naïke Bochatay.

**Writing – review & editing:** Nadia M. Bajwa, Julia Sader, Sara Kim, Yoon Soo Park, Mathieu R. Nendaz, Naïke Bochatay.

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
