## [Decision Letter · Decision Letter 0]

21 Sep 2022

PONE-D-22-19898Development and validity evidence for the Intraprofessional Conflict Exercise: An assessment tool to support collaborationPLOS ONE

Dear Dr. Bajwa,

Thank you for submitting your manuscript to PLOS ONE. After careful consideration, we feel that it has merit but does not fully meet PLOS ONE’s publication criteria as it currently stands. Therefore, we invite you to submit a revised version of the manuscript that addresses the points raised during the review process.

We look forward to receiving your revised manuscript.

Kind regards,

Conor Gilligan

Academic Editor

PLOS ONE

Journal Requirements:

Additional Editor Comments:

The reviewers have provided useful feedback and some clear direction for you to improve the clarity of your manuscript if you wish to revise and resubmit. Please address the reviewers' comments in a revised version.

Reviewers' comments:

Reviewer's Responses to Questions

**Comments to the Author**

1. Is the manuscript technically sound, and do the data support the conclusions?

Reviewer #1: Partly

Reviewer #2: Yes

2. Has the statistical analysis been performed appropriately and rigorously? 

Reviewer #1: I Don't Know

Reviewer #2: I Don't Know

3. Have the authors made all data underlying the findings in their manuscript fully available?

Reviewer #1: Yes

Reviewer #2: Yes

4. Is the manuscript presented in an intelligible fashion and written in standard English?

Reviewer #1: Yes

Reviewer #2: Yes

5. Review Comments to the Author

Reviewer #1: Thank you for the opportunity of reviewing this interesting article that explores intraprofessional conflict management as an important aspect of collaborative healthcare practice. The Introduction and Discussion draw on key literature. While the article is well written, making socio-cultural aspects of the Interprofessional Conflict Exercise (ICE) more explicit in the Method will strengthen its argument and usefulness.

Introduction: Conflict management in the Introduction is located within the complex intersection of healthcare, collaboration and education. The authors clearly highlight the richness of this complex intersection through their consideration of interprofessional collaboration, intraprofessional collaboration, curricula and assessment; in relation to communication, relationships, negotiation, interpersonal factors, power, social identify theory, context and feedback.

Methods: A socio-cultural perspective on this complex space is indicated by the literature drawn on in the Introduction, the inclusion of a sociologist in the development of the tool and the use of qualitative data. Accordingly, there is scope to relate the development of ICE more explicitly back to the complex space. This could be done by explaining to the reader the rationale behind:

- the focus on post-registration for conflict management (rather than pre-registration);

- the use of a professional development scenario (that is preparation for a journal club rather than a patient care situation);

- the framing of ICE as an exercise (rather than as assessment for an admissions process); and

- not referring to any previous training participants may have had, or not had, for communication, negotiation and assertiveness skills (including whether this could have been explicit or implicit in pre- or post-registration curricula and where this might be and what it might look like).

There is also scope for the Methods to provide more details about:

- the sources of literature accessed in developing the ICE (including whether this was limited to literature related to intraprofessional collaboration in health, or whether it included literature related to communication, professionalism, leadership, management and beyond);

- the assessors’ training (including whether the training was at familiarising assessors with the process of using ICE and/or calibrating assessors and/or developing assessors’ understandings of the components of ICE);

- qualitative researchers training (and if not done, why not);

- how what was ‘expected’ and what was deemed ‘appropriate’ were determined (and if this was in relation to participants’ level of training or what was required in the interview and how consensus was reached);

- whether there was a debrief following the simulated exercise; and

- whether the score was shared with participants (including whether it was the basis of a reflective discussion or not).

Results: The use of both quantitative and qualitative data is a strength of the study. The qualitative data provides a depth of understanding and a clear illustration of the use of the ICE, thus facilitating readers’ decisions about the transferability of the ICE to their own contexts. I will not comment on the statistical analysis.

Discussion: The Discussion would benefit from authors’ reflections on the implications of:

- generalisability or transferability of ICE to other professions (based on medicine being the discipline of focus, and including the characteristics of the medical profession that will influence this facilitate or hinder this);

- generalisability or transferability of ICE beyond the specific scenario for which it was developed;

- generalisability or transferability of ICE as an assessment tool for interviews, versus as the basis for an exercise for developing conflict management;

- actual or potential overlaps with other educational areas focusing on communication, recognition of limits and demonstration of respect (including whether these could be for example, complementary or synergistic);

- the relationship between education for intraprofessional collaboration and silo-based education.

Reviewer #2: The manuscript is well written, particularly the introduction. The topic is highly relevant given the need for valid tools to assess conflict and interpersonal interactions in healthcare professions.

The following limitations were identified:

1) The title is somehow misleading, because the manuscript does not report any intervention or study on how the tool developed can support collaboration.

2) P. 4, references 20-21: the example given seems to more relevant for inter-professional than for intraprofessional conflict (i.e. social identity theory).

3) The authors searched literature on intraprofessional conflict and developed the tool based on the literature identified. However, it is not clear if the literature search was performed specifically for healthcare (as mentioned in the abstract) or more generally. On page 7, the authors may specify this important aspect in the text. It would also be relevant to explain in more details to what extent intraprofessional conflict is different or similar to conflict in other work domains.

4) The simulated task is not a typical “medical” situation. The preparation of a journal club may not reflect the reality of many clinical situations where intraprofessional conflicts may be observed. The authors may discuss this aspect as major limitation and specify to what extent conflict management skills in a simulated non –medical situation (i.e. the journal club) may reflect conflict management skills is clinical situations, or not.

5) The authors may specify to what extent the rating system bases on occurrences (i.e. frequency) or quality of the items that were observed.

6) In the data collection section, some important aspects need to be clarified: What did the applicants were told before the situation? Did they receive a feedback on their own performance? Were the raters present in the room or were also video recordings used during the study? What did the faculty examiners vs sociologists/psychologists did? How was the tool used in practice (e.g. discussion or reflection after the situation?).

7) A schematic representation of the development of the tool may help the readers to quickly understand the different steps the authors took to develop the tool.

6. PLOS authors have the option to publish the peer review history of their article (what does this mean?). If published, this will include your full peer review and any attached files.

Reviewer #1: No

Reviewer #2: No

---

## [Author Response · Author response to Decision Letter 0]

31 Oct 2022

October 12th, 2022

Dear Prof. Gilligan, 

Thank you so much for your careful read of our manuscript (PONE-D-22-19898), “Development and validity evidence for the Intraprofessional Conflict Exercise: An assessment tool to support collaboration.” 

My co-authors and I have carefully reviewed and made every effort to address each concern in this revision. 

Again, we are very pleased to have the opportunity to share our findings with the health sciences community through publication in PLOS ONE. We thank you for considering our manuscript and await your reply. 

Sincerely yours,

Nadia Bajwa, MD, MHPE, PhD

Médecin Adjointe, Pédiatrie Générale, Responsable de la Formation Post-graduée, HUG

Membre facultaire, Unité de Développement et de Recherche en Education Médicale (UDREM), UNIGE

Département de la femme, de l'enfant et de l'adolescent

6, rue Willy-Donzé - 1211 Genève 14

Tél. +41 22 37 23082, +41 79 55 33705

Email: nadia.bajwa@hcuge.ch

Manuscript ID#: PONE-D-22-1989

Manuscript Title: Development and validity evidence for the Intraprofessional Conflict Exercise: An assessment tool to support collaboration

Journal Requirements:

->We have revised the naming of the supporting information to meet PLOS ONE requirements.

->The data in this study has been deposited in the repository Yareta. It may be found at this link: https://doi.org/10.26037/yareta:ar2yfvbqfrhn7pvzlilno4uvym

There are no restrictions on the data.

->The ethics statement is now exclusively in the Methods section and has been removed from the end of the manuscript.

Additional Editor Comments:

The reviewers have provided useful feedback and some clear direction for you to improve the clarity of your manuscript if you wish to revise and resubmit. Please address the reviewers' comments in a revised version.

Reviewers' comments:

Reviewer's Responses to Questions

Comments to the Author

1. Is the manuscript technically sound, and do the data support the conclusions?

Reviewer #1: Partly

Reviewer #2: Yes

2. Has the statistical analysis been performed appropriately and rigorously?

Reviewer #1: I Don't Know

Reviewer #2: I Don't Know

3. Have the authors made all data underlying the findings in their manuscript fully available?

Reviewer #1: Yes

Reviewer #2: Yes

4. Is the manuscript presented in an intelligible fashion and written in standard English?

Reviewer #1: Yes

Reviewer #2: Yes

5. Review Comments to the Author

Reviewer #1: Thank you for the opportunity of reviewing this interesting article that explores intraprofessional conflict management as an important aspect of collaborative healthcare practice. The Introduction and Discussion draw on key literature. While the article is well written, making socio-cultural aspects of the Interprofessional Conflict Exercise (ICE) more explicit in the Method will strengthen its argument and usefulness.

Introduction: Conflict management in the Introduction is located within the complex intersection of healthcare, collaboration and education. The authors clearly highlight the richness of this complex intersection through their consideration of interprofessional collaboration, intraprofessional collaboration, curricula and assessment; in relation to communication, relationships, negotiation, interpersonal factors, power, social identify theory, context and feedback.

-> We thank reviewer one for their kind words. We have made every effort to respond to reviewer one’s comments below.

Methods: A socio-cultural perspective on this complex space is indicated by the literature drawn on in the Introduction, the inclusion of a sociologist in the development of the tool and the use of qualitative data. Accordingly, there is scope to relate the development of ICE more explicitly back to the complex space. This could be done by explaining to the reader the rationale behind:

- the focus on post-registration for conflict management (rather than pre-registration);

->As requested by reviewer one, we have now made explicit our focus on post-graduate students in the methods. “We chose to have the case take place in the post-graduate setting to simulate a workplace setting.”

We also clarified that the ICE can be used at any stage of health professions education: “The use of the ICE between undergraduate and graduate medical education was appropriate given applicants’ likely pervious exposure to training in communication. However, the tool may be used at any stage of the health professions education continuum.”

- the use of a professional development scenario (that is preparation for a journal club rather than a patient care situation);

->We have now made explicit our reason for not including a patient care situation in the manuscript. “A patient care situation was not included in the case as the two other cases already involved patient care situations and because intraprofessional conflicts can arise for various reasons beyond patient care disagreements [18].”

- the framing of ICE as an exercise (rather than as assessment for an admissions process); and

->We thank reviewer 1 for pointing this out and we have now modified the methods to indicate that we intend for the ICE to be used in settings other than the admissions process. “While the use of the ICE is limited in our study to a residency admissions process, the ICE is intended to be used in a variety of settings as part of conflict management training.”

- not referring to any previous training participants may have had, or not had, for communication, negotiation and assertiveness skills (including whether this could have been explicit or implicit in pre- or post-registration curricula and where this might be and what it might look like).

->As requested by reviewer one, we have now included information on prior communication and conflict management training of the participants. “Participants were not questioned on their previous communication skills or conflict management training. Most applicants completed medical school at the University of Geneva where communication skills training includes but is not limited to conducting an interview with a patient, breaking bad news, and conducting handoffs. Conflict management is not specifically addressed. The use of the ICE between undergraduate and graduate medical education was appropriate given applicants’ likely previous exposure to training in communication. However, the tool may be used at any stage of the health professions education continuum.”

There is also scope for the Methods to provide more details about:

- the sources of literature accessed in developing the ICE (including whether this was limited to literature related to intraprofessional collaboration in health, or whether it included literature related to communication, professionalism, leadership, management and beyond);

->The sources of literature are mentioned in the manuscript: “The 22 item ICE is based on the concepts of principled negotiation, the DESC method (describe, express, specify, and consequences), the principles of non-violent communication, the CanMEDS collaborator role, and the Professionalism Mini-Evaluation exercise 1-4.”

- the assessors’ training (including whether the training was at familiarising assessors with the process of using ICE and/or calibrating assessors and/or developing assessors’ understandings of the components of ICE);

->We have now elaborated on the examiner training for the ICE. “Two trained pediatric faculty examiners observed and rated each applicant using the ICE. We trained examiners to use the ICE prior to the simulation by having them observe a video recording of the SC case and by practicing filling in the instrument. Examiners were specifically briefed on their understanding of the ICE items. A discussion with the examiners was used to develop consensus about what an appropriate response to the situation would be.”

- qualitative researchers training (and if not done, why not);

->We have now added this to the methods. “As the qualitative researchers were involved in the creation of the ICE, they did not undergo rater training.”

- how what was ‘expected’ and what was deemed ‘appropriate’ were determined (and if this was in relation to participants’ level of training or what was required in the interview and how consensus was reached);

->This information has now been added to the manuscript. “A discussion with the examiners was used to develop consensus about what an appropriate response to the situation would be.”

- whether there was a debrief following the simulated exercise; and

- whether the score was shared with participants (including whether it was the basis of a reflective discussion or not).

-> This information has now been added to the methods. “Participants were not debriefed immediately after the simulated exercise and scores were not immediately shared with the participants. Participants were invited after the admissions decision to a voluntary debriefing session where a video of the performance was viewed, scores were shared, and constructive feedback was given to each participant.”

Results: The use of both quantitative and qualitative data is a strength of the study. The qualitative data provides a depth of understanding and a clear illustration of the use of the ICE, thus facilitating readers’ decisions about the transferability of the ICE to their own contexts. I will not comment on the statistical analysis.

->We thank reviewer one for their feedback.

Discussion: The Discussion would benefit from authors’ reflections on the implications of:

- generalisability or transferability of ICE to other professions (based on medicine being the discipline of focus, and including the characteristics of the medical profession that will influence this facilitate or hinder this);

-> We thanks reviewer one for their remark. We have now modified the discussion to address generalizability of the ICE: “Other health professions such as nursing share contextual and cultural similarities to physicians. We believe that the items in the ICE may be generalizable to other health professions and that future studies testing the ICE with other health professions may contribute to the literature on intraprofessional conflict.”

- generalisability or transferability of ICE beyond the specific scenario for which it was developed;

-> We have now added to the limitations the use of the ICE with only one scenario: “We used the ICE with one case scenario focused on a professional development task. The study of the use of the ICE with various intraprofessional collaboration challenges, specifically those concerning patient care, would add to the ICE’s validity evidence.”

- generalisability or transferability of ICE as an assessment tool for interviews, versus as the basis for an exercise for developing conflict management;

->We have now modified the discussion to highlight the transferability of the ICE to conflict management curricula: “A strength of our study is the reliable performance of the ICE in a high-stakes assessment such as an admissions process. The established validity evidence lays the groundwork for and supports future use of the ICE as a formative assessment in a conflict management curriculum.”

- actual or potential overlaps with other educational areas focusing on communication, recognition of limits and demonstration of respect (including whether these could be for example, complementary or synergistic);

->We have now added to the discussion a mention of other educational areas: “The three domains of the ICE (communication skills, recognition of limits, and demonstration of respect for others) may already be taught in other communication or teamwork curricula; However, contextualizing the use of these skills in the setting of conflict allows for a synergistic application of these skills.”

- the relationship between education for intraprofessional collaboration and silo-based education.

->We have now made this distinction in the discussion: “The advantage of using the ICE in simulation is that it allows learners to step out from a theoretical silo where responses to situations are predictable and permits them to respond to complex situations in a safe environment.”

Reviewer #2: The manuscript is well written, particularly the introduction. The topic is highly relevant given the need for valid tools to assess conflict and interpersonal interactions in healthcare professions.

->We thank reviewer two for their kind words. We have done our best to respond to the identified limitations.

The following limitations were identified:

1) The title is somehow misleading, because the manuscript does not report any intervention or study on how the tool developed can support collaboration.

->We have now modified the discussion to focus more on collaboration. We hope now that the use of the ICE to support collaboration is more explicit and that the title is indicative of this purpose.

2) P. 4, references 20-21: the example given seems to more relevant for inter-professional than for intraprofessional conflict (i.e. social identity theory).

-> We have now added an example to the introduction to make more explicit the relationship between intraprofessional conflict and social identity theory: “For example, physicians who identify themselves as belonging to one subspecialty may look down upon physicians who work in primary care.”

3) The authors searched literature on intraprofessional conflict and developed the tool based on the literature identified. However, it is not clear if the literature search was performed specifically for healthcare (as mentioned in the abstract) or more generally. On page 7, the authors may specify this important aspect in the text. It would also be relevant to explain in more details to what extent intraprofessional conflict is different or similar to conflict in other work domains.

->As our literature search went beyond the domain of healthcare, we have removed this qualifying phrase from the abstract. We have also broadened our description of the literature search in the methods: “The 22 item ICE is based on our review of the literature on conflict management in health care and beyond, including the concepts of principled negotiation, the DESC method (describe, express, specify, and consequences), the principles of non-violent communication, the CanMEDS collaborator role, and the Professionalism Mini-Evaluation exercise 1-4.”

4) The simulated task is not a typical “medical” situation. The preparation of a journal club may not reflect the reality of many clinical situations where intraprofessional conflicts may be observed. The authors may discuss this aspect as major limitation and specify to what extent conflict management skills in a simulated non –medical situation (i.e. the journal club) may reflect conflict management skills is clinical situations, or not.

->This same point was also mentioned by reviewer one. We have now modified the limitations to mention this point: “We used the ICE with one case scenario focused on a professional development task. The study of the use of the ICE with various intraprofessional collaboration challenges, specifically those concerning patient care, would add to the ICE’s validity evidence.”

5) The authors may specify to what extent the rating system bases on occurrences (i.e. frequency) or quality of the items that were observed.

->We thank reviewer two for asking for this clarification. We have now modified the methods to make this point more explicit: “For each item, the participant’s overall performance is rated. Frequency counts of each behavior are not meant to be collected.”

6) In the data collection section, some important aspects need to be clarified: What did the applicants were told before the situation? Did they receive a feedback on their own performance? Were the raters present in the room or were also video recordings used during the study? What did the faculty examiners vs sociologists/psychologists did? How was the tool used in practice (e.g. discussion or reflection after the situation?).

->We have now added these details to the methods concerning the information given to participants beforehand and the process for giving feedback afterwards: “Before entering the simulation, participants were introduced to the scenario with a written introduction giving context to the case and specific tasks to be performed (review the instructions for the journal club with their colleague, choose an article to be presented together, and decide on the distribution of labor for the preparation of the presentation). Participants were not debriefed immediately after the simulated exercise and scores were not immediately shared with the participants. Participants were invited after the admissions decision to a voluntary debriefing session where a video of the performance was viewed, scores were shared, and constructive feedback was given to each participant.”

->We have also now specified the positioning of the raters: “On the day of the simulation, raters observed each participant through a two-way mirror.”

->The role of the psychologist and sociologist is specified in the methods: “On the day of the simulation, one of two qualitative researchers (a sociologist or a psychologist, depending on the day) observed the SC encounters. As the qualitative researchers were involved in the creation of the ICE, they did not undergo rater training. They used the ICE as a semi-structured observation guide and took notes on how each item manifested itself in the encounter. They also paid particular attention to interactions and processes at play during the simulated conflict.”

7) A schematic representation of the development of the tool may help the readers to quickly understand the different steps the authors took to develop the tool.

->We have now added a schematic representation of the development of the tool to the supporting information: “A schematic representation of the ICE development process is described in S2 Fig.” 

6. PLOS authors have the option to publish the peer review history of their article (what does this mean?). If published, this will include your full peer review and any attached files.

Do you want your identity to be public for this peer review? For information about this choice, including consent withdrawal, please see our Privacy Policy.

Reviewer #1: No

Reviewer #2: No

1. Berger E, Chan M-K, Kuper A, et al. The CanMEDS role of Collaborator: How is it taught and assessed according to faculty and residents? Paediatr Child Health. 2012;17(10):557-560. doi:10.1093/pch/17.10.557

2. Bower SA, Bower GH. Asserting Yourself-Updated Edition: A Practical Guide For Positive Change. Hachette UK; 2009.

3. Cruess R, McIlroy JH, Cruess S, Ginsburg S, Steinert Y. The Professionalism Mini-Evaluation Exercise: A Preliminary Investigation. Acad Med. 2006;81(Suppl):S74-S78. doi:10.1097/00001888-200610001-00019

4. Rosenberg MB, Chopra D. Nonviolent communication: A language of life: Life-changing tools for healthy relationships. PuddleDancer Press; 2015.

---

## [Decision Letter · Decision Letter 1]

8 Dec 2022

PONE-D-22-19898R1Development and validity evidence for the Intraprofessional Conflict Exercise: An assessment tool to support collaborationPLOS ONE

Dear Dr. Bajwa,

Thank you for submitting your manuscript to PLOS ONE. After careful consideration, we feel that it has merit but does not fully meet PLOS ONE’s publication criteria as it currently stands. Therefore, we invite you to submit a revised version of the manuscript that addresses the points raised during the review process.

Myself as well as the original reviewer 1 have reviewed your revised manuscript and we are both pleased with the extent to which you have addressed the earlier comments. I have a couple of remaining queries/concerns that I invite you to address before submitting a final version.

We look forward to receiving your revised manuscript.

Kind regards,

Conor Gilligan

Academic Editor

PLOS ONE

Journal Requirements:

Additional Editor Comments:

Thank you for responding to the reviewers' earlier comments. I can see that you have comprehensively done so. In the absence of a second reviewer for this revision I have reviewed the manuscript and am happy to say that pending a few minor changes I think it will soon be ready to be accepted for publication.

Please consider:

1. The information you have added on the bottom of page 6 (to address reviewer queries about the setting, transferability of findings, previous communication skills training and application in different stages of health professions education) seems to be a little clumsy in that it is all together and some of it somewhat out of place. I would suggest that most parts of this would be best placed in the limitations section in the discussion, with the exception of the reference to application in undergraduate vs postgraduate training.

2. I am still not entirely clear on the justification for not trialing this with undergraduate students – using a workplace scenario makes sense but why not test this with students to see how it works as a teaching tool before they enter the workplace? The goals and conclusions all point to application for medical student training but the ICE has not been tested in that context.

Typo:

-Abstract - final sentence of methods and findings section - there is a repeat word 'in'

Reviewers' comments:

Reviewer's Responses to Questions

**Comments to the Author**

1. If the authors have adequately addressed your comments raised in a previous round of review and you feel that this manuscript is now acceptable for publication, you may indicate that here to bypass the “Comments to the Author” section, enter your conflict of interest statement in the “Confidential to Editor” section, and submit your "Accept" recommendation.

Reviewer #1: All comments have been addressed

2. Is the manuscript technically sound, and do the data support the conclusions?

Reviewer #1: Yes

3. Has the statistical analysis been performed appropriately and rigorously? 

Reviewer #1: I Don't Know

4. Have the authors made all data underlying the findings in their manuscript fully available?

Reviewer #1: Yes

5. Is the manuscript presented in an intelligible fashion and written in standard English?

Reviewer #1: Yes

6. Review Comments to the Author

Reviewer #1: The authors have addressed all my comments. The article makes a valuable contribution to the literature addressing the complexity of and education for healthcare practice.

7. PLOS authors have the option to publish the peer review history of their article (what does this mean?). If published, this will include your full peer review and any attached files.

Reviewer #1: No

---

## [Author Response · Author response to Decision Letter 1]

12 Dec 2022

December 12th, 2022

Dear Prof. Gilligan, 

We would like to thank you for your further consideration of our manuscript (PONE-D-22-19898), “Development and validity evidence for the Intraprofessional Conflict Exercise: An assessment tool to support collaboration.” 

In this second revision, my co-authors and I have carefully reviewed and made every effort to address each concern in this revision. 

Again, we are very pleased to have the opportunity to share our findings with the health sciences community through publication in PLOS ONE. We thank you for considering our manuscript and await your reply. 

Sincerely yours,

Nadia Bajwa, MD, MHPE, PhD

Médecin Adjointe, Pédiatrie Générale, Responsable de la Formation Post-graduée, HUG

Membre facultaire, Unité de Développement et de Recherche en Education Médicale (UDREM), UNIGE

Département de la femme, de l'enfant et de l'adolescent

6, rue Willy-Donzé - 1211 Genève 14

Tél. +41 22 37 23082, +41 79 55 33705

Email: nadia.bajwa@hcuge.ch

Manuscript ID#: PONE-D-22-1989

Manuscript Title: Development and validity evidence for the Intraprofessional Conflict Exercise: An assessment tool to support collaboration

Journal Requirements:

-> All references have been reviewed and verified to comply with the PLOS One style.

Additional Editor Comments:

Thank you for responding to the reviewers' earlier comments. I can see that you have comprehensively done so. In the absence of a second reviewer for this revision I have reviewed the manuscript and am happy to say that pending a few minor changes I think it will soon be ready to be accepted for publication.

Please consider:

1. The information you have added on the bottom of page 6 (to address reviewer queries about the setting, transferability of findings, previous communication skills training and application in different stages of health professions education) seems to be a little clumsy in that it is all together and some of it somewhat out of place. I would suggest that most parts of this would be best placed in the limitations section in the discussion, with the exception of the reference to application in undergraduate vs postgraduate training.

->We thank the Editor for their comment. We have now reformulated the section on page 6 and have also moved the discussion on prior communication skills training to the limitations in the discussion:

“Most applicants completed medical school at the University of Geneva where communication skills training includes but is not limited to conducting an interview with a patient, breaking bad news, and conducting handoffs. Conflict management is not specifically addressed. The use of the ICE between undergraduate and graduate medical education was appropriate given applicants’ likely previous exposure to training in communication. While the use of the ICE is limited in our study to a residency admissions process, the ICE is intended to be used in a variety of settings as part of conflict management training at any stage of the health professions education continuum.”

Limitations:

“In addition, participants were not questioned on their previous communication skills or conflict management training and this may have had an impact on their performance.”

2. I am still not entirely clear on the justification for not trialing this with undergraduate students – using a workplace scenario makes sense but why not test this with students to see how it works as a teaching tool before they enter the workplace? The goals and conclusions all point to application for medical student training but the ICE has not been tested in that context.

-> We thank the editor for their valid comment. We have now added this perspective to our limitations:

“The study of the use of the ICE with various intraprofessional collaboration challenges, specifically those concerning patient care, would add to the ICE’s validity evidence. More specifically, we would like to see the application of the ICE in an undergraduate teaching setting on conflict management with the goal of preparing students for the workplace.”

Typo:

-Abstract - final sentence of methods and findings section - there is a repeat word 'in'

->We thank the editor for their careful read and have made the correction.

Reviewers' comments:

Reviewer's Responses to Questions

Comments to the Author

1. If the authors have adequately addressed your comments raised in a previous round of review and you feel that this manuscript is now acceptable for publication, you may indicate that here to bypass the “Comments to the Author” section, enter your conflict of interest statement in the “Confidential to Editor” section, and submit your "Accept" recommendation.

Reviewer #1: All comments have been addressed

2. Is the manuscript technically sound, and do the data support the conclusions?

Reviewer #1: Yes

3. Has the statistical analysis been performed appropriately and rigorously?

Reviewer #1: I Don't Know

4. Have the authors made all data underlying the findings in their manuscript fully available?

Reviewer #1: Yes

5. Is the manuscript presented in an intelligible fashion and written in standard English?

Reviewer #1: Yes

6. Review Comments to the Author

Reviewer #1: The authors have addressed all my comments. The article makes a valuable contribution to the literature addressing the complexity of and education for healthcare practice.

7. PLOS authors have the option to publish the peer review history of their article (what does this mean?). If published, this will include your full peer review and any attached files.

Do you want your identity to be public for this peer review? For information about this choice, including consent withdrawal, please see our Privacy Policy.

Reviewer #1: No

---

## [Editor Report · Decision Letter 2]

2 Jan 2023

Development and validity evidence for the Intraprofessional Conflict Exercise: An assessment tool to support collaboration

PONE-D-22-19898R2

Dear Dr. Bajwa,

We’re pleased to inform you that your manuscript has been judged scientifically suitable for publication and will be formally accepted for publication once it meets all outstanding technical requirements.

Kind regards,

Nabeel Al-Yateem, PhD

Academic Editor

PLOS ONE
---

## [Editor Report · Acceptance letter]

6 Jan 2023

PONE-D-22-19898R2 

Development and validity evidence for the Intraprofessional Conflict Exercise: An assessment tool to support collaboration 

Dear Dr. Bajwa:

I'm pleased to inform you that your manuscript has been deemed suitable for publication in PLOS ONE. Congratulations! Your manuscript is now with our production department. 

Kind regards, 

on behalf of

Dr. Nabeel Al-Yateem 

Academic Editor

PLOS ONE